# An Overview on Cognitive Function Enhancement through Physical Exercises

**DOI:** 10.3390/brainsci11101289

**Published:** 2021-09-29

**Authors:** Narayanasamy Sai Srinivas, Vijayaragavan Vimalan, Parasuraman Padmanabhan, Balázs Gulyás

**Affiliations:** 1Lee Kong Chian School of Medicine, Translational Neuroscience, 59 Nanyang Drive, Nanyang Technological University, Singapore 636921, Singapore; saivicky93@gmail.com (N.S.S.); vimalan.vijay@ntu.edu.sg (V.V.); balazs.gulyas@ntu.edu.sg (B.G.); 2Cognitive Neuroimaging Centre, 59 Nanyang Drive, Nanyang Technological University, Singapore 636921, Singapore; 3Imaging Probe Development Platform, 59 Nanyang Drive, Nanyang Technological University, Singapore 636921, Singapore; 4Department of Clinical Neuroscience, Karolinska Institute, 17176 Stockholm, Sweden

**Keywords:** cognitive function, physical activity, exercise, imaging modalities, fMRI, fNIRS, EEG

## Abstract

This review is extensively focused on the enhancement of cognitive functions while performing physical exercises categorized into cardiovascular exercises, resistance training, martial arts, racquet sports, dancing and mind-body exercises. Imaging modalities, viz. functional magnetic resonance imaging (fMRI), functional near-infrared spectroscopy (fNIRS) and electroencephalography (EEG), have been included in this review. This review indicates that differences are present in cognitive functioning while changing the type of physical activity performed. This study concludes that employing fNIRS helps overcome certain limitations of fMRI. Further, the effects of physical activity on a diverse variety of the population, from active children to the old people, are discussed.

## 1. Introduction

Apart from physical development, cognitive development is an important domain of human growth. Human cognition comprises large-scale networks that are functionally coherent at rest and collectively active during cognitive processes [1]. Cognition alludes to “the psychological activity or process of gaining information and comprehension through idea, experience, and the sense”. It has been branched into nonsocial and social cognition. Nonsocial cognition refers to the mental abilities of an individual, such as his or her attention span, processing speed, problem solving and reasoning skills, as well as working memory. The psychological processes involved in perception, encoding, storage, retrieval and control of knowledge about oneself and others are collectively labeled as social cognition [2]. Cognitive efficiency refers to the complex construct that represents the capacity to achieve learning and problem-solving skills through the optimal usage of mental resources [3]. Many mental health disorders have been observed recently during childhood and adolescence, whereby cognition is highly associated with the quality of life of the individual. At time, the social, mental and biological components decide a person’s emotional well-being, and these can largely influence a person’s quality of life. We also studied the cognitive abilities of stroke and schizophrenia patients in this review. The lifestyles and difficulties of stroke patients were evaluated with simple reaction time tests (cognitive tests) and quality of life assessments. The attention and visuospatial skills are highly associated with the timing of a stroke, where the reaction time tasks act as a marker for quality of life [4]. Cognitive impairment decreases the quality of life over a period of 12 months from the time of diagnosis. Cognitive impairments were also set as a diagnostic criterion for schizophrenia, ranging from moderate to severe during the first outbreak and persisting throughout the period of illness. They have been shown to negatively affect the adaptive life skills of the patients during the later period of their lives [5]. Focusing the four realms of social cognition has been a core intervention in treating schizophrenia, which includes emotional processing, theory of mind (ToM), attributional bias, social perception and social knowledge. Early clinical interventions are required to provide a positive outcome [6]. Recent research on social cognitive processing is used to analyze various disorders such as anxiety, eating and mood disorders among clinical and non-clinical samples [7].

Several scientific approaches have been adapted in enhancing the performance of human cognition [8]. While most of the strategies for enhancing cognition rely on pharmacological, environmental and genetic factors [9], an embodied approach may serve as a better alternative. Such an embodied system for cognition opens the opportunity for various disciplines of science, technology, education and research to incorporate learning tools to upgrade the pedagogy [10]. Such a type of training incorporates perceptual motor activities and cognitive challenges tailored to individual needs as per their respective goals [11]. Among the factors that were examined, stress may act as a facilitating or debilitating element for cognitive function, based on variables such as magnitude, cause (due to external factors or the inherent task) and duration (acute or chronic). Sandi et.al. demonstrated that while mild stress enhances cognitive function in times of easier activities, high stress levels lead to decrements regardless of the cause [12]. Despite sparse studies, observational data has supported that tiredness, lack of attention, easy irritability, loss of sleep and an inability to manage emotions, leading to a decline in executive functioning, may be dictated by the prefrontal cortex [13]. It is noteworthy that a correlation between sleep and cognition has been well documented. In children, loss of sleep results in a decline in cognitive functions.

Reversibility was shown to be dependent on baseline cognitive functioning and involvement in social activities. In analyzing the reversion of mild cognitive impairment, higher chances of recovery were observed midlife (52%) in comparison with late life (26.6%) among a Korean community. Therefore, early interventions are more effective [14] for treating cognitive impairment. Genetics has been shown to be a major determinant for cognitive ability. However, genes encoding, dystrobrevin-binding protein 1 (DTNBP1), brain-derived neurotrophic factor (BDNF), catechol-O-methyl transferase (COMT) and apolipoprotein E (APOE), have demonstrated a repeated association with cognitive decline [15]. Age, education, gender, occupation and lifestyle factors such as smoking also impact cognitive functioning. In addition, the environment at home and around family members plays an important role (Figure 1) [16]. Green environments have also shown enhancement in cognition [17]. Nutritional intake has demonstrated effects on cognition in children and adolescents [18].

Cognitive dysfunction has great clinical significance and longstanding ramifications on a patient’s activities. Pharmacological (e.g., antidepressants) and non-pharmacological (e.g., cognitive-oriented treatments and cognitive remediation therapy (CRT)) interventions focusing predominantly on learning and memory, which potentially improve the cognitive depression of the brain, are being advanced. Non-pharmacological interventions employ cognition remediation therapy and cognitive-oriented treatments, with increased cognition in patients with depression [20]. Particular strategies have been used to promote memory, problem-solving and decision-making abilities [21]. In one study, cognitive training was employed as an intervention to enhance driving skills in elderly people. Five weeks of training was provided to enhance aspects such as attention control along with memory and processing speed. This was shown to be more relevant in the elderly population, with a major contribution to improvement of the working memory [22], since working memory forms a vital component of cognition that is effective during inhibition and switching processes, which are dependent on the maturation of the prefrontal lobes. Performing interventions in the earlier stages of childhood as described earlier in the text has shown better impacts in enhancing the functions of the prefrontal lobes, as they are not yet fully developed in children. Major techniques involve using language as a mediator for enhancement in cognition control [23]. Different categories of cognitive interventions, namely cognitive stimulation (CS), focusing on social abilities and cognition; cognitive training (CT), or the repeated practice of tasks performed in daily life; cognitive rehabilitation (CR), a personalized approach toward impaired functions; and a hybrid of CT and CS (MCTS) were employed in 33 studies for dementia patients. Assessments were made using mini-mental state examination (MMSE) and the Alzheimer’s Disease Assessment Scale-Cognition (ASAS-cog). While CS showed positive enhancements in cognition, CR, CT and MCTS did not demonstrate any significant changes [24].

Physical exercise has also been found to play a pivotal role in enhancing cognitive function for all age groups, where it elevates the levels of cerebral blood flow and growth factors, including brain-derived neurotrophic factor and neurotransmitters (e.g., dopamine and norepinephrine) [25]. Furthermore, physical exercise has served as an intervention for the aging population to enhance cognitive functioning and prevent cognitive decline [26]. Exercise has been shown to be effective in making notable lifestyle changes in the elderly population and minimizing the risk of neurological disorders such as Alzheimer’s disease [27]. While physical activity has shown increments in academic performance, it occurred mainly in females. This study highlights the need to conduct further experiments to understand the role of sex and the intensity of exercise along with the psychological variables [28]. An integration of physical activity and cognitive tasks has been established in multiple studies. Additionally, there has been an analysis of the variations of physical activities and how it impacts the cognitive state. A 20-week physical exercise program for an elderly population with moderate cognitive impairment has resulted in post–intervention improvements [29], whereas a 12-week physical activity program led to an enhancement in cognitive function and brain activation among the elderly population when combined with cognitive exercises. Clinical research has demonstrated a abundant hippocampal and basal ganglia volume, greater white matter integrity, efficient brain activity along with superior cognitive performance and academic achievements for physically active and fit children and preadolescents. Physical fitness and regular exercise have been shown to positively impact the prefrontal cortex, functional brain connectivity and executive and memory functions [30]. The impact of physical exercises has been explored in detail in the following section.

## 2. Types of Physical Activities That Enhance Cognitive Functions

### 2.1. Aerobic Exercise

Performing aerobic exercise for a period of 20 min demonstrated enhanced cognitive function among preadolescents and young adults. In this study, acute exercise was performed in a single bout at a moderate intensity for 20 min on a treadmill, with a 65–75% heart rate reserve along with a five minutes warm up and cool down [31]. A separate study also demonstrated an increase in cognitive functioning and working memory due to better cortical activity after acute aerobic exercise for 20 min with a cycling ergometer among female college students [32]. Twenty sedentary young adults performed 30 min of aerobic exercise on a stationary bike. This demonstrated an increase in activation of the left dorsolateral prefrontal cortex and left orbital frontal cortex. The control group, which did not perform exercise, did not see any benefits, but an enhanced performance in cognitive tasks was seen in the exercising group [33]. Better enhancements in cognition were seen when aerobic training was combined with CRT. BDNF, being a pivotal element for brain development and growth, has shown increased levels after aerobic exercise. However, further studies are required to identify other factors [34].

Functional MRI assessment demonstrated better cognitive control among children through aerobic fitness enhancement [35]. Regular aerobic exercise has been shown to reduce neuronal decline in the elderly population. Furthermore, it acts as an efficient mechanism for the restoration of normal functions due to changes in the brain’s structure [36]. As discussed earlier in the text, performing aerobic exercises demonstrated to be beneficial in enhancing the cognitive capabilities of the elderly population, especially in the regions of driving memory, inhibition of redundant information and switching between tasks. This was achieved predominantly by the frontal lobes of the brain. Capillary development in the brain and an increase in the length and number of dendritic synapses between neurons were among the processes involved [37]. It has been discovered that the impacts of aerobic exercises on cognition and executive functioning are due to the influence of the temporal, frontal and parietal brain regions. The frontal lobe performs a significant part in maintaining executive functioning in aging. Meanwhile, performing regular aerobic exercise showed an impact on the hippocampal region (Figure 2) [38]. Enhancements in neural efficiency were observed in seventeen volunteers with mild cognitive impairment following a twelve week exercise intervention of supervised treadmill walking. This led to an increase in cognition mainly due to the activation of 11 brain regions—especially in older adults—and has been proven effective for delaying Alzheimer’s disease [39].

### 2.2. Resistance Training

Resistance training has gained popularity in recent years. While it was popular only among bodybuilders and powerlifters earlier, lately it has found fresh interest among athletes in other fields, such as soccer and basketball. Recently, it has also been followed closely by several fitness enthusiasts who aim to achieve benefits such as muscular mass growth and strength. Moreover, the contribution of such training to cognitive development should not be dismissed. A study comparing the effects of 30 min of aerobics and resistance training in a group of untrained youths reported similar increases in cognition in comparison with the control group [40]. Positive correlation has been established between cortical hemodynamics of the prefrontal cortex and the handgrip strength of the individual. This was analyzed among 39 young adults aged 18–30 who used a handgrip dynamometer to assess strength and fNIRS for identifying the cortical hemodynamics of the prefrontal cortex [41].

Incorporating resistance training at a frequency of twice a week in addition to aerobic exercise and balance training for a period of 52 weeks for a group of 155 elderly women showed enhancements in memory and a drop in cortical white matter size (Figure 3) [42]. Upon reviewing multiple studies of resistance training in healthy elders, cognitive differences were observed. Resistance training protocols mixed with chess and gambling can limit cognitive decline and enhance the QoL, including attention, calculation, recall and language. Resistance training potentially elevates the level of insulin-like growth factor 1 (IGF-1). IGF-1 raises the synthesis of BDNF1 and vascular endothelial growth factor (VEGF), which improves cognitive functioning [43]. A 6-month resistance training intervention was performed on aged persons with type-2 diabetes who were also at increased risk of cognitive deterioration. Resistance training’s effects on cognition have also been compared with water-based exercise among older adults. A mixture of free weights and HUR equipment operated by pressurized air weights were used to target all the primary muscle groups, such as quadriceps, pectoralis major and latissimus dorsi. Assessments of maximal muscular strength were performed after the third week. The control group solely performed basic exercises like stretching and bodyweight exercises. However, the group that performed resistance training demonstrated better increments in cognition and brain health, hence illustrating this as a cost-effective intervention for the aging population [44]. An elderly group of 100 (aged 70 years) was subjected to a six month intervention and random assignment to cognitive training or a mix of both of cognitive and resistance exercise. Resistance training was proven to be superior for improving global cognition and expanding gray matter volume in the posterior cingulate rather than cognitive training alone. Cognitive training was beneficial to memory due to greater functional communication between the hippocampus and superior frontal cortex [45]. Similar results of cognitive enhancements were found in both groups in comparison to the control group, except for the reaction time. The water-based exercise group had better results in that domain [46].

### 2.3. Sports

Martial arts and combat sports have brought about many benefits to cognitive health. While most forms of such sports originated in East Asia geographically, it has been popularized worldwide. In modern times, many forms of martial arts are viewed as a sport and are attributed to many health benefits including strength, power, endurance and balance [47]. Martial arts have also been studied as a means to develop cognitive functioning, given its predisposition to mind-body nature. The complex movements have a greater cortical requirement in comparison with aerobic exercises or resistance training. Certain elements of cognition, such as concentration and mindfulness, which may not be adequately developed in aerobic or resistance training sessions are enhanced in martial arts sessions. An increased regional cerebral blood flow has also been observed following more complex moves, enhancing executive functioning, which serves as a pivotal component in cognition, while aerobic exercises could only improve attention and the processing speed [48]. Magnetic resonance imaging (MRI) assessments were performed to compare 14 professional karateka and healthy controls. All the professionals demonstrated higher cognitive performances alongside improved motor performances due to the demands of the sport [49]. In another study, karate instruction was imparted to 39 sedentary children to analyze its effects on motor and cognitive development. A wide range of cognitive functions saw improvement in those who had practiced karate, which was assessed by the visual selective attention task, tasks measuring working memory function and the Tower of London test [50]. Three months of karate training had significantly benefited the cognitive functions of healthy older adults, mainly pertaining to visual tasks, executive functions, motion tasks and memory. The main components of karate are basic techniques, kata (forms) and combat practice. Assessments were performed through various questionnaires that included a memory-complaint battery of tests and MMSE [51].

In another study comprising aged subjects, an enhancement of cognition was observed in the aspects of attention, resilience and motor reaction time following a 5-month intervention with 89 older adults aged above 70 years. The main reason was attributed to the combination of aerobic, balance and coordination functions found in this East Asian martial art [52]. In another study, 1 h of weekly taekwondo training was provided to a group of 24 healthy volunteers aged above 40 years. The intervention lasted for a period of 4 months. It comprised the components of a warm-up, muscle strengthening and stretching, followed by taekwondo-based techniques. These components were aimed at improving stances, blocking, kicking, punching and the usage of kicking pads. All of them had shown improvement in reaction time and motor timing [53]. Taekwondo was analyzed for its efficiency in implementation in a public school setting. The executive functioning skills of the students who took part in taekwondo were higher than the control group. Computer administration was employed to assess executive functioning [54]. Around 30 healthy men volunteered to analyze the effects of the Japanese martial art kendo. Kendo practitioners demonstrated higher functional connectivity, mainly between the left intraparietal sulcus and left precentral gyrus [55]. The attention network test was performed on 48 participants of 2 groups, in which one had martial arts experience and the other did not. It was seen that all the martial artists had higher working memory, attention and behavioral inhibition. While this was more common among children, adults had improved corticospinal excitability resulting from karate and better motor cortex function from taekwondo. It was also seen that most benefits, especially among older adults, were due to chronic adaptations and not acute ones [56]. When analyzing the brain connectivity in children, taekwondo was proven to be effective in enhancing functional interconnection from the cerebellum to the inferior frontal gyrus. They also had enhanced minimal frequency oscillations in the right frontal precentral gyrus and right parietal precuneus. This resulted in enhanced intelligence [19].

Higher inhibitory control (the capacity to suppress planned but unsuitable preparatory activities in the current environment) have been vital in human performance. Sports requiring open skills such as tennis have been more advantageous in enhancing this component of cognition in humans in comparison with closed skill sports such as swimming. Tennis players also demonstrate faster reactions with better accuracy during cognitive tasks. This type of skill training has been shown to benefit inhibitory control when combined with aerobic training instead of aerobic training alone [57]. It was observed that tennis was effective in enhancing action observation, anticipation and motor control through increased activation of the cerebellum and action observation network (AON) [58]. Another experiment analyzed tennis with fNIRS and observed an increase in the left premotor regions, which led to enhanced cognition, especially in unpredictable conditions [59]. To differentiate elite table tennis players and non-players, fMRI was employed. While the players had around 8 years of experience, the non-players had no prior experience. Diminished cortical activity was observed in the athletic group in the context of sports-related and unrelated visual-spatial tasks. Shorter reaction times were also seen among the players, leading to faster responses, which are indicative of higher neural efficiency. The areas studied included the right supplementary motor area, right angular gyrus, left supramarginal gyrus and right paracentral lobule. However, it must be noted that this adaptation is an effect of long-term training (Figure 4) [60]. Similar to tennis, recreational badminton players demonstrated higher anticipation due to the open skill nature of the sport, which involved the cortical areas pertaining to observation and interpretation of others’ actions. Enhancements were mainly noticed in the medial, dorsolateral and ventrolateral frontal cortices [61].

Higher anticipation rates were observed among 15 basketball players in comparison with the novices. The athletic group also displayed more accuracy for anticipation along with gaze behavior. The attribution was mainly due to the increased activity in the inferior parietal lobe and inferior frontal gyrus (Figure 5) [62]. In a study of 21 basketball players, higher portion of gray matter, mainly in the left anterior insula, inferior frontal gyrus, inferior parietal lobule and right anterior cingulate cortex, and increased functional connectivity in the resting state were observed in comparison with the novices. These adaptations enhance the executive control network, leading to better cognition (Table 1) [63].

### 2.4. Dance

Apart from musical experience, which has been shown to provide a positive influence on the health of the participant, the effects of dance have also been studied. When analyzing 475 amateur dancers, benefits were seen in the physical as well as social dimensions. This has raised the importance of dance as a medium in health benefits [65]. Introducing dance as an exercise among older adults showed greater improvements in brain volumes in comparison with the studies discussed in the previous section, which focused on aerobic exercises, anaerobic strength and stretching exercises. Enhanced brain volumes from the 6-month intervention caused higher cognitive functions, such as working memory and attention hence attenuating the age-related cognitive decline. This could be attributed to the multi-faceted nature of dance, encompassing spatial orientation, movement coordination, balance, endurance, interaction and communication (Figure 6) [64].

Activation of the medial superior parietal lobule and improvement in the proprioceptive and somatosensory aspects were caused by the spatial navigation of leg movements during dance and controlled muscle contractions [67]. Six months of dance intervention was performed for a cohort of seniors. There were noticeable changes in terms of attention and cognition between the intervention and control groups. Improvements were seen in the corticospinal tracts in the group that underwent intervention [68].

### 2.5. Yoga

Yoga is a popular kind of mind-body workout which consists of meditation, breathing exercises and posture control. Systematic investigations have been made on its physical and psychological health benefits, and it serves as an intervention for circumstances like as arthritis, pain and musculoskeletal conditions. Certain psychological conditions such as depression and anxiety have been reverted through yoga. Over the years, its effects on enhancing cognition have been an area of active research to identify the underlying neurological correlates. The gray matter volume was higher in the left hippocampus of experienced yoga practitioners, and the reduced activity in the dorsolateral prefrontal cortex compared with the controls. Reaction times were similar in both groups [69]. Yoga has demonstrated greater activation of the prefrontal cortex, and practitioners exhibit higher executive functions independent of negative emotional stimuli [66]. Performing yoga has been displayed as a successful strategy for enhancing mindfulness, leading to prevention of age-related decline in fluid intelligence along with enhanced brain functioning [70].

## 3. Neuroimaging Modalities

Functional neuroimaging techniques have aided the examination of human brain function and created a revolution in neuroscience studies. Applications of these modalities have helped to assess the brain damage and to understand how the surviving brain networks are altered, which aids potentially to improve the treatment [71]. These techniques have demonstrated a wide range of diverse applications, including understanding neuropsychiatric illnesses such as obsessive-compulsive disorder [72], the neural substrates of parent–infant attachment [73] and gait disorders [74]. Neuroimaging has also been employed in analyzing the advantages of physical activity on brain health and cognitive functioning. It has shown to be an effective tool in understanding both the structural and functional elements of cognitive processes [75]. It has been shown to be effective in measuring many dimensions such as the gray matter volumes [76], cortical regions which perform an important role in episodic, spatial memory and impacted cognitive performance [77] and hippocampal volume [36]. A few key neuroimaging modalities used in examining brain function are explored below.

### 3.1. Functional Magnetic Resonance Imaging (fMRI)

fMRI is a popular neuroimaging technique which has facilitated several developments pertaining to medical applications, mainly for the human brain. Its advantages include the non-invasive nature of the machine and noise reduction. Applications include both the diagnosis and treatment of brain lesions [78]. In cognitive neuroscience, fMRI is important since it is employed to generate pictures of brain functions. These are important in many aspects of neuroscience, such as understanding language, social interactions of the individual, behavior and cognition [79]. It was also shown to be effective in assessing pain elicited by noxious heat in 14 healthy participants [80]. Recent improvements in the fMRI provided the ability to present high-definition images of the brain while performing various motor tasks. This has aided in comprehending various structures of the human brain, which can be affected while performing physical activity. However, constraints have existed due to the low magnetic properties and limitations in head movement while performing exercises. Due to this, predominantly only small muscle groups, such as those for performing handgrip strength, have worked in this modality, but studies have been made on cycle ergometers [81]. This method also solved a research gap of identifying the neurobiological pathways implicated in exercise-induced alterations during cognitive processing. It demonstrated that pathway variation between the elderly and younger individuals. While in the elderly, alterations were limited to the left parietal lobe, widespread changes were observed in the right hemisphere for the younger age group [82]. This method was also used to investigate and demonstrate concurrent and independent connections between different dimensions of physical fitness, working memory, and brain function in over 1000 healthy people. [83]. The major advantage of fMRI is the spatial resolution and whole-brain coverage. Considering its limited temporal resolution, correlations between multiple brain regions and temporal delays of the brain might be performed [84].

### 3.2. Functional near Infrared Spectroscopy (fNIRS)

Functional near infrared spectroscopy (fNIRS) has gained popularity and rapidly advanced over the past decade in the cognitive neuroscience field. This can primarily be attributed to its advantages over other neuroimaging devices such as fMRI and EEG in terms of its safety, tolerance to bodily movement, portability and suitability for all age groups, ranging from newborns to the elderly. It has aided in the comprehension of different aspects of cognition which were not adequately demonstrated in other modalities. Those areas include functional specialization in the visual and sensorimotor systems, face and language processing and in assessing tasks performed in everyday life [85]. fNIRS has been shown to be an effective neuroimaging modality for assessing the effects of exercise on cerebral oxygenation and hemodynamics due to its capability in quantifying changes (Figure 7) [86].

It has helped for comparing the effectiveness of high-intensity aerobic exercises with moderate-intensity exercises in enhancing working memory and neural activity in patients dependent on methamphetamine (MA). The former was proven to be more effective and has been established as a therapeutic tool [87]. The neural bases of cognitive contributions in the gait were also investigated with fNIRS to record brain activation. Simple and complex walking could be assessed in adults of different age groups and those with balance disorders. Greater prefrontal cortical activation was seen in those who performed secondary tasks such as arithmetic along with walking [88]. Application of this modality was extended to understanding the cognition and attentional demands required by lower limb amputees. The amputees demonstrated similar cognitive strategies in challenging situations but had increased attention demands in regular walking [89]. Tai Chi Chuan (TCC), a popular mind body exercise involving cognitive training and movement meditations, was also examined. The effect of TCC on enhancing the brain’s prefrontal structure function and memory was observed via fMRI. However, fNIRS would be a suitable imaging modality, which measures the change in the HbO_2_ and HHb levels for the tasks involving body movements compared with other noninvasive neuroimaging techniques [90]. The major disadvantage of this modality for clinical use is its lack of accuracy and precision. The employment of high-density whole head optode arrays, precise sensor locations relative to the head, short-distance channels, anatomical co-registration and multi-dimensional signal processing could be performed to enhance the sensitivity of fNIRS. This could make it a widespread clinical tool for assessing brain function [91].

### 3.3. Electroencephalography (EEG)

Electroencephalography (EEG) measures the electric potentials of the brain and helps for investigating the cortical activity in lab and field settings. Its cost-effectiveness and compact size provide an advantage over other modalities such as functional magnetic resonance imaging (fMRI). Its high temporal resolution offers precise means for examining the cognitive processes [92]. EEG was employed in identifying the relationship between skipping and cognitive learning. Three-minute bouts of skipping had shown increments in both repetitive and variable learning [93]. Weight training was also an interest for identifying its effects on the brain and cognition. A common exercise, the bench press, was performed, and EEG served as an efficient tool for recording brain amplitudes. Significant increments were observed in the beta and gamma frequency bands. However, further studies are required to assess the changes with respect to the intensity and its effects on cognition and perception [94]. A 12-week resistance training program was performed among elderly subjects with mild cognitive impairment with elastic bands. Fifteen repetitions were employed with 65% 1RM for all the exercises. Increments in cognitive functioning were observed, mainly for the alpha and theta power [95]. The effects of yoga on brain waves were also analyzed with EEG. Positive effects from performing this mind-body exercise for enhancing cognition, along with alleviation of blood pressure and anxiety, were seen while using EEG [96]. Despite the importance in the application of representing the neuronal activity of the brain, many applications fail to take use of all of the information accessible from the brain’s dynamic sources. Localization of the sources of the brain’s signals is pivotal for any analysis [97].

## 4. Conclusions

This study looked at the effect of physical activity on enhancing cognitive performance in a variety of individuals, ranging from children to the elderly. The mechanism of the increase in cognitive function could be attributed to the increased hippocampal and basal ganglia volume and greater white matter integrity. Enhanced cerebral flow along with alterations in neurotransmitter release and structural changes in the central nervous system have also shown they can serve as physiological mechanisms to explain its effect. Comparisons in improvements have been made between aerobic exercises, resistance training, mind-body exercises such as yoga, racket sports such as tennis and basketball, combat sports and dance. While performing regular aerobic exercises like cycling has shown benefits, resistance training was proven to be more effective in improving certain dimensions of cognition to a greater degree than aerobic exercises. These exercises have helped via enhanced functioning of the BDNF, where it has been able to stop the neuronal decline caused by age and aid in the growth of capillaries in the brain. Major benefits included enhancements in memory and changes in brain volume. Involvement in open skill sports such as martial arts and tennis has demonstrated greater corticospinal excitability, motor cortex function, faster reaction times with better accuracy and better inhibitory control in comparison with performing aerobic exercises or resistance training. The modalities employed were fMRI and fNIRS. Even though the former was helpful for identifying brain images and functioning, there were certain limitations, such as low magnetic properties and limitation in head movement while performing the exercises. fNIRS helped to overcome these limitations though its safety, portability and suitability. It could be understood that variations in physical activity have shown different effects on cognitive functioning. More studies are required to understand the best choice of exercise for improving cognition.

## Figures and Tables

**Figure 1 brainsci-11-01289-f001:**
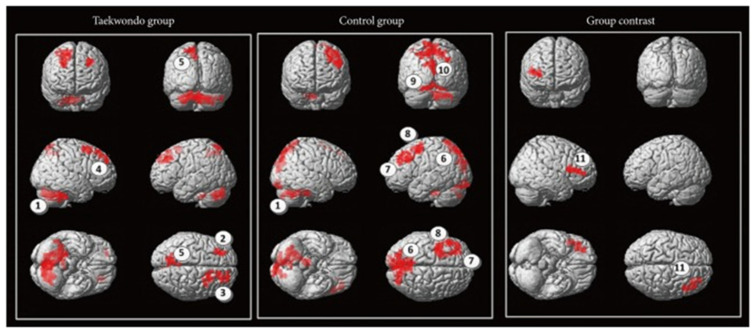
Taekwondo group demonstrating enhanced functional connectivity (Reprinted with permission [19]).

**Figure 2 brainsci-11-01289-f002:**
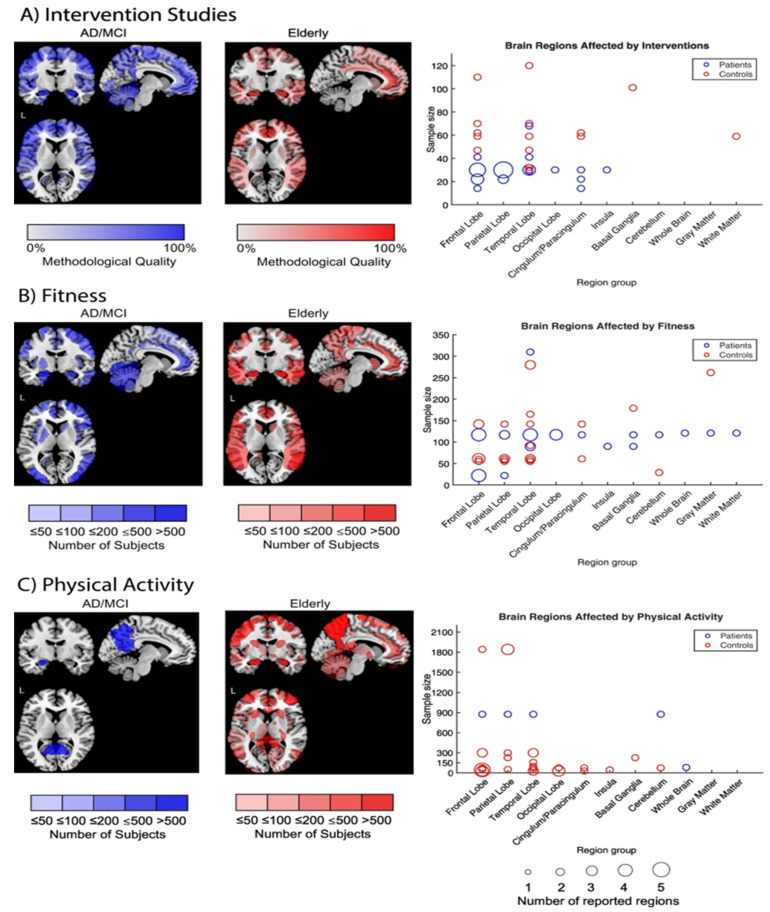
Overview of the brain regions affected by the intervention studies (**A**) intervention studies, (**B**) Fitness, (**C**) Physical activity. Blue figures indicate healthy adults, and red figures refer to the elderly. Comparisons were made between the volume and cortical thickness (Reprinted with permission [38]).

**Figure 3 brainsci-11-01289-f003:**
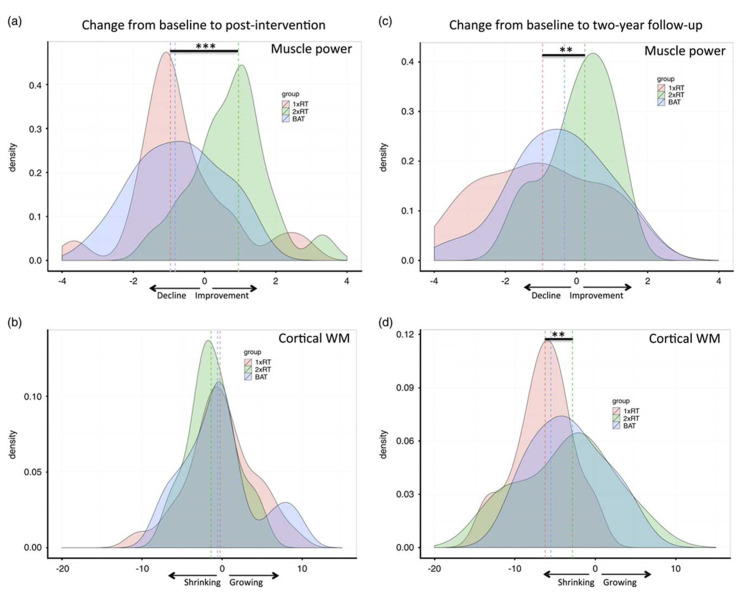
Impact of treatment conditions on change in peak muscle power and white matter volume from baseline to post-intervention (**a**,**b**) and from baseline to 2-year follow-up (**c**,**d**). Peak muscle power (in watts) values have been divided by 100. Cortical white matter volumes are expressed in cm^3^, and the dashed vertical lines represent the median score for each distribution. ** *p* < 0.01. *** *p* < 0.001. (Used with permission [42]).

**Figure 4 brainsci-11-01289-f004:**
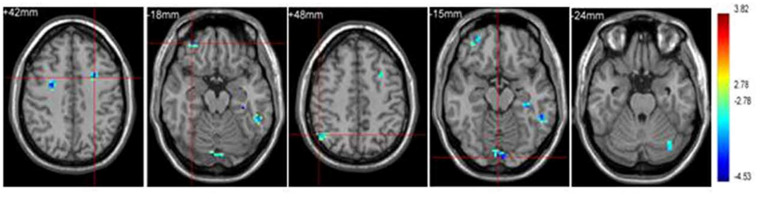
The brain regions (right supplementary motor area, right paracentral lobule, left supramarginal gyrus and right angular gyrus) that were activated for the table tennis players (Used with permission [60]).

**Figure 5 brainsci-11-01289-f005:**
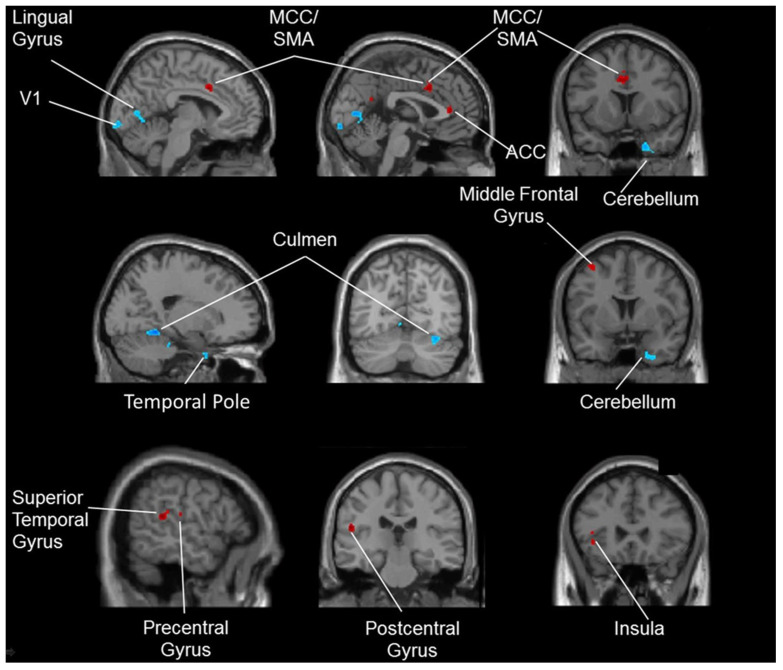
Higher activity was observed in the inferior parietal lobule and frontal gyrus in athletes compared with novices (Used with permission [64]).

**Figure 6 brainsci-11-01289-f006:**
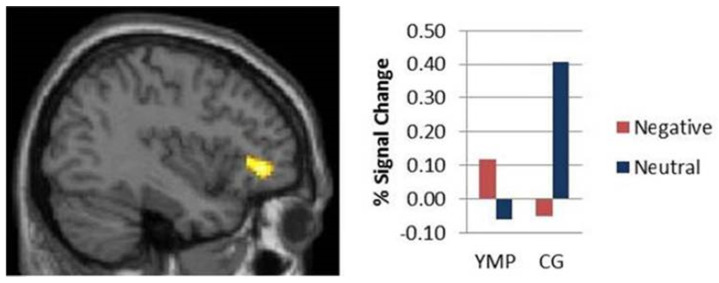
The dance group demonstrated higher volumes in the frontal and temporal cortical areas compared with the sports group. Red-colored regions provide an overview of the regions ([66] with permission).

**Figure 7 brainsci-11-01289-f007:**
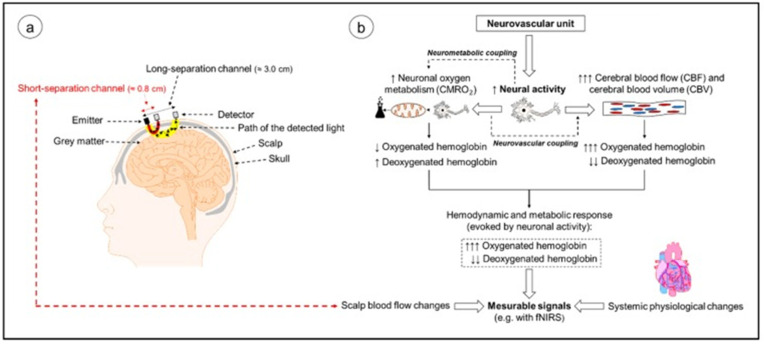
(**a**) Neural activity-induced oxygenation, changes in neurovascular units and changes in cerebral hemodynamics. (**b**) An illustration of NIRS on the human head and assumed shape of light and “long separation channels” (Used with permission [86] ).

**Table 1 brainsci-11-01289-t001:** Larger gray matter volumes were observed in basketball players in multiple areas, including the right precuneus, left anterior insula, right anterior cingulate cortex, left inferior frontal gyrus and left inferior parietal lobule (Used with permission [63]).

Brain Region	Side	x	y	z	Voxels	*t*-Value
Precuneus (BA 31)	R	11	−74	23	52	8.52
Anterior insula (BA 13)	L	−42	−6	−0	136	7.88
Anterior cingulate cortex (BA 32)	R	2	32	6	35	6.32
Inferior frontal gyrus (BA 9)	L	−45	11	33	48	6.75
Inferior parietal lobule (BA 3)	L	−45	−24	44	75	7.50

BA: Brodmann’s area; L: left; R: right. Coordinates refer to Talairach space. Brain areas with corrected *p* < 0.05 were listed.

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
