# Peer review of "An Overview on Cognitive Function Enhancement through Physical Exercises"

_brainsci, 2021, doi:10.3390/brainsci11101289_

Round 1

Reviewer 1 Report

In this manuscript, the authors provided a review of studies on the cognitive enhancement effects of physical exercise. The topic is an interesting one and has received much attention recently. However, the manuscript is poorly written and I recommend rejection.

1), many of the arguments throughout the manuscript are confusing, poorly organized, conceptually inaccurate, lack focus, and lack proper citation. For instance, the 1st paragraph emphasizes human development and cognitive development, while the 2nd paragraph emphasizes quality of life, what is the connection?

The definition of cognition (line 28) is outdated and different from mainstream cognitive psychology.

Line 31-33: whose quality of life? what is "the reaction time"?

In the 2nd paragraph, the authors first bring about the topic of quality of life, then discuss social cognition, then go back to human cognition again. What is the relation among these concepts?

In lines 39-40, the authors stated that emotional processing and attributional bias are social cognition, which is basically wrong because they largely involve nonsocial cognition.

Lines 41-43, lines 53-54, lines 119-121, lines 212-215: what do these sentences mean? Lines 41-43,  lines 119-121: references missed.

Lines 178-180 vs lines 500-501: inconsistent conclusions.

The way how the references are cited is inappropriate throughout the manuscript. It is mainly because the authors summarize the findings of each study in several sentences and only add the references to the last sentence; the expressions are also confusing such that it is hard to comprehend the first sentence, how it is linked to the previous and subsequent arguments on the same study.

2), as a review paper on physical exercise, it is also unclear why the authors spent more than one page on other factors that affect cognition.

3), each section lacks overall guiding statements that help the readers comprehend what the authors want to say and show.

4), section 3 Neuroimaging modalities, this is a main part of the review, however, this part is somewhat superficial, without providing any insightful discussions on the common and unique findings of each technique and how the field can be advanced.

Author Response

Replies to reviewers:

Thank you to the reviewers for taking their precious time off to assess our manuscript.

Reviewer -1

1), many of the arguments throughout the manuscript are confusing, poorly organized, conceptually inaccurate, lack focus, and lack proper citation. For instance, the 1st paragraph emphasizes human development and cognitive development, while the 2nd paragraph emphasizes quality of life, what is the connection?

The definition of cognition (line 28) is outdated and different from mainstream cognitive psychology.

The definition for the Cognition is updated (Line no: 30-32)

Line 31-33: whose quality of life? what is "the reaction time"?

We have included additional points for the above queries. Whose Quality of life– the lifestyle and difficulties faced among the stroke patients after they encounter stroke. Reaction time – speed of processing the given task which displays the quality of life of the patients. (Line no: 33-40)

In the 2nd paragraph, the authors first bring about the topic of quality of life, then discuss social cognition, then go back to human cognition again. What is the relation among these concepts?

The relationship between the topics is discussed in the manuscript (Line No: 33-41)

Cognition had been branched as nonsocial and social cognition. Non-social cognition refers to the mental abilities of the individual such as the attention span, processing speed, problem solving, reasoning and working memory.

  The psychological processes involved with the perception, encoding, storage, retrieval and regulation of information about self and others are collectively labelled social cognition. (Green et al., 2019) 

Targeting the four domains of social cognition has been a core intervention in treating schizophrenia, which includes emotional processing, Theory of Mind (ToM), attributional bias, social perception and social knowledge. This was quoted in (Javed A, Charles A. The importance of social cognition in improving functional outcomes in schizophrenia. Frontiers in psychiatry. 2018;9:157.) 

In lines 39-40, the authors stated that emotional processing and attributional bias are social cognition, which is basically wrong because they largely involve nonsocial cognition.

This was quoted in (Javed A, Charles A. The importance of social cognition in improving functional outcomes in schizophrenia. Frontiers in psychiatry. 2018;9:157.) 

Lines 41-43, lines 53-54, lines 119-121, lines 212-215: what do these sentences mean? Lines 41-43,  lines 119-121: references missed.

We corrected them as per the reviewer’s comments

Lines 178-180 vs lines 500-501: inconsistent conclusions.

The inconsistent conclusions lines were corrected

The way how the references are cited is inappropriate throughout the manuscript. It is mainly because the authors summarize the findings of each study in several sentences and only add the references to the last sentence; the expressions are also confusing such that it is hard to comprehend the first sentence, how it is linked to the previous and subsequent arguments on the same study.

Thank you very much for the suggestions! We carried out all corrections mentioned by the reviewer

2), as a review paper on physical exercise, it is also unclear why the authors spent more than one page on other factors that affect cognition.

We removed the unwanted text from the descriptions and the running text is more appropriate.

3), each section lacks overall guiding statements that help the readers comprehend what the authors want to say and show.

Thank you very much for the reviewer the valuable comments! We completely changed to a meaningful text

4), section 3 Neuroimaging modalities, this is a main part of the review, however, this part is somewhat superficial, without providing any insightful discussions on the common and unique findings of each technique and how the field can be advanced.

Thank you very much! We corrected this section in the text.

Reviewer 2 Report

The manuscript provides an interesting overview of the role different types of physical activity play in enhancing human cognition. Some of the written text is not clear or not very succinct and suggestions are made for improvement. As this is an ‘overview’ more could be made to enhance the limitations of the methods used or when one method may be more appropriately used compared with another.

Author Response

Reviewer -2

Thank you to the reviewers for taking their precious time off to assess our manuscript.

The manuscript provides an interesting overview of the role different types of physical activity play in enhancing human cognition. Some of the written text is not clear or not very succinct and suggestions are made for improvement. As this is an ‘overview’ more could be made to enhance the limitations of the methods used or when one method may be more appropriately used compared with another.

Immense thanks for your suggestion for the improvement of the manuscript!  We did corrections as suggested by the reviewer, formatted, and highlighted according to the guideline of the Journal. 

Round 2

Reviewer 1 Report

Most of my concerns are not well addressed, the definitions of relevant concepts are inaccurate, and I still cannot see how the manuscript contributes to the field. The manuscript needs a comprehensive brushup by a senior researcher preferably a psychologist.

Author Response

Thank you to the reviewers for taking their precious time off to assess our manuscript.

Reviewer round 2-

Most of my concerns are not well addressed, the definitions of relevant concepts are inaccurate, and I still cannot see how the manuscript contributes to the field. The manuscript needs a comprehensive brushup by a senior researcher preferably a psychologist.

The manuscript was corrected by a senior professor and the reviewer comments in round 1 are answered below.

Reviewer Round 1

1), many of the arguments throughout the manuscript are confusing, poorly organized, conceptually inaccurate, lack focus, and lack proper citation. For instance, the 1st paragraph emphasizes human development and cognitive development, while the 2nd paragraph emphasizes quality of life, what is the connection?

Many mental health disorders are arising during childhood and adolescence where the cognition is highly associated with the quality of life of the individual. At some random time, social, mental, and biological components decide a people's emotional well-being, and this can influence a people's quality of life (Line no 33-36)

The definition of cognition (line 28) is outdated and different from mainstream cognitive psychology.

Cognition alludes to "the psychological activity or process of gaining information and comprehension through idea, experience, and the sense” (Line no: 30-31)

Line 31-33: whose quality of life? what is "the reaction time"?

The lifestyle and difficulties of the stroke patients have been evaluated with the simple and choice reaction time tests (cognitive tests), and assessment of quality of life. The attention and visuospatial skills are highly associated with stroke where the reaction time tasks are act as a marker for quality of life. (Line no: 44-47)

In the 2nd paragraph, the authors first bring about the topic of quality of life, then discuss social cognition, then go back to human cognition again. What is the relation among these concepts?

The 1st and 2nd paragraphs have been altered for better understanding (Line no: 39-59).  

In lines 39-40, the authors stated that emotional processing and attributional bias are social cognition, which is basically wrong because they largely involve nonsocial cognition.

Please refer the paper Javed A, Charles A. The importance of social cognition in improving functional outcomes in schizophrenia. Frontiers in psychiatry. 2018;9:157 where the above statement quoted

Lines 41-43, lines 53-54, lines 119-121, lines 212-215: what do these sentences mean? Lines 41-43, lines 119-121: references missed.

  1. 41-43: The sentence dealing with recent research findings on the social cognitive processing to analyze various disorder which is cited in the text reference number 7.

Recent research on social cognitive processing is used to analyze the various disorders such as anxiety, eating and moods disorders between clinical and non-clinical samples Line No (56-58)

  1. 53-54: The sentence discusses the embodied system of cognition in upgrading pedagogy (the sentence modified for better understanding). whereas an embodied approach may serve as a better alternative. These embodied system for cognition gives a chance to science, technology, education, and mathematics disciplines to incorporate embodied learning tools to upgrade pedagogy (Line no: 69-72).
  2. 119-121: The role of exercise on cognitive improvement has been discussed and sentence rewritten for better understanding which is cited in the text reference number 24

Physical exercise has also been found to play a pivotal role in enhancing cognitive function where it elevates the levels of cerebral blood flow, growth factors (brain-derived neurotrophic factor and neurotransmitters (dopamine and norepinephrine) (Line No:140-142).

  1. 212-215: The sentence has been modified.

Resistance training protocols mixed with chess gambling can limit the cognitive decline and enhance the QoL such as attention, calculation, recall and language etc., Resistance training is potentially increasing the level of insulin-like growth factor 1 (IGF-1). IGF-1 elevates the production of brain derived neural facctor-1 (BDNF1) and vascular endothelial growth factor (VEGF), which may improve the cognitive functions (Line no: 246-251)

Lines 178-180 vs lines 500-501: inconsistent conclusions.

The inconsistent conclusion lines 500-501 were deleted (Line no 554-555)

The way how the references are cited is inappropriate throughout the manuscript. It is mainly because the authors summarize the findings of each study in several sentences and only add the references to the last sentence; the expressions are also confusing such that it is hard to comprehend the first sentence, how it is linked to the previous and subsequent arguments on the same study.

The reference sections have been changed throughout the manuscript and some of the sentences rewritten for better understanding.

2), as a review paper on physical exercise, it is also unclear why the authors spent more than one page on other factors that affect cognition.

We removed the unwanted text from the descriptions and the running text is more appropriate.

3), each section lacks overall guiding statements that help the readers comprehend what the authors want to say and show.

We have summarized the overall of guiding statements with few lines at the front of each section, which may help to readers to understand what we may going to explain in the section.

4), section 3 Neuroimaging modalities, this is a main part of the review, however, this part is somewhat superficial, without providing any insightful discussions on the common and unique findings of each technique and how the field can be advanced.

The common and unique feature for each technique was added in the manuscript

Line no: 477-480; 518-423; 543-546

Reviewer 2 Report

The authors failed to implement changes in my first detailed feedback to them. There has been some changes in this revised manuscript. The authors do need to go through the text of the manuscript more carefully to enhance understanding for the reader. I would get a native reader of English to check the written text. There are sentences that are not completely clear.

Author Response

Thank you to the reviewers for taking their precious time off to assess our manuscript.

Round 2

The authors failed to implement changes in my first detailed feedback to them. There have been some changes in this revised manuscript. The authors do need to go through the text of the manuscript more carefully to enhance understanding for the reader. I would get a native reader of English to check the written text. There are sentences that are not completely clear.

The manuscript was carefully read by the senior professor and the language has been improved in the text.

Round 1

The manuscript provides an interesting overview of the role different types of physical activity play in enhancing human cognition. Some of the written text is not clear or not very succinct and suggestions are made for improvement. As this is an ‘overview’ more could be made to enhance the limitations of the methods used or when one method may be more appropriately used compared with another.

We have improved the text throughout the manuscript for better understanding. Regarding methods, the advantage and/or disadvantage has been added to each section where the readers can understand the limitations of the instruments.

Line no: 477-480; 518-423; 543-546